# Are You Dominated by Your Affects? How and When Do Employees’ Daily Affective States Impact Learning from Project Failure?

**DOI:** 10.3390/bs13060514

**Published:** 2023-06-19

**Authors:** Wenzhou Wang, Longdi Li, Shanghao Song, Wendi Jiang

**Affiliations:** 1Business School, Beijing Normal University, Beijing 100875, China; wenzhou.wang@mail.bnu.edu.cn; 2School of Public Administration, Renmin University of China, Beijing 100872, China; lld0531@ruc.edu.cn; 3School of Government, Beijing Normal University, Beijing 100875, China; shanghaosong@mail.bnu.edu.cn

**Keywords:** positive affective states, negative affective states, error management strategy, project commitment, learning from failure

## Abstract

Given the enormous value that project failure brings to individuals and organizations, a large number of scholars have explored the antecedents that affect employees’ learning from project failure. However, few scholars have paid attention to how individuals’ affective states interact with cognition patterns to learn from failure. Based on cognitive behavioral theory, this paper explores the relationship between employees’ different daily affective states and learning from project failure and incorporates the mediating role of error management strategy and the moderating role of project commitment. By using SPSS and Amos software, hierarchical regression analysis of questionnaire data collected from 774 employees in high-tech firms in China indicates that (1) positive/negative affective states positively/negatively affect learning from failure, respectively; (2) error management strategy mediates the relationship between daily affective states and learning from project failure; and (3) project commitment moderates the relationship between negative affective states and error management strategy; specifically, this relationship is weaker when project commitment is stronger. However, the moderating effect of project commitment on the relationship between positive affective states and error management strategy is not supported. The results further expand the research related to learning from failure and have practical implications for failure management in high-tech enterprises.

## 1. Introduction

In today’s turbulent society, especially since the start of the COVID-19 pandemic, many SMEs are in an existential crisis and are experiencing more failures than successes [1]. In previous studies, failure is considered part of an important “learning journey” [2], which even provides more valuable information than the results of success or failure themselves. It requires individuals to take time to learn from their failures, which is critical to their later “turning defeat into victory” [3]. The essence of learning from failure lies in solving potential or existing problems. We should, thus, not only take measures to achieve expected results, but also investigate the root causes of problems to understand why they occurred, and finally learn from them to avoid repeating them in the future [4].

Usually, individuals show general daily affective states at work, and different affective states usually bring about different behavioral consequences. General daily affective states can be both positive and negative. Some scholars have linked emotions or affective states to learning behavior to explore the close relationship between the two of them [5,6,7], believing that positive emotions or affective states can promote learning, while the negative will hinder learning. Specifically, positive affective states can push individuals to overcome a psychological dilemma after a failure faster and help them analyze problems and learn from failure [8]. In contrast, negative affective states at work can have a significant negative impact on the learning process and learning outcomes [9]. However, most of these studies have been conducted by psychologists or educationists, and scholars in the field of organization have not yet paid attention to the impact of employees’ daily affective states on learning behavior. In addition, as a special situational event, few scholars have focused on the relationship between an individual’s affective states and learning behavior after experiencing the failure, or suggested appropriate mediating mechanism and boundary conditions. The relationship between daily affective states and learning from failure has been mostly viewed as common knowledge. Therefore, we intend to add a more comprehensive set of antecedents to previous studies to investigate the specific mechanisms by which general daily affective states affect learning from failure.

Although scholars have mentioned that an individual’s affective state affects learning behavior, the specific mechanism involved is not clear [10]. In the process of learning from project failure, an individual’s attitude and strategy for dealing with errors is crucial. An error management strategy is a strategy for individuals to actively think about errors and seek solutions [11]. It can minimize the negative outcomes and maximizes the positive effects of errors [12]. Error management strategy can enable individuals to focus on the positive aspects of past failures and view failure as an opportunity to learn. Such individuals can then effectively deal with errors that have occurred, mitigate their negative consequences, and avoid future errors [13,14]. We argue that employees’ daily affective states can influence the error management process by influencing the process of acquiring resources necessary for error management and employees’ attitudes towards failure, and help employees seize learning opportunities and improve their learning ability, and ultimately have a positive impact through learning from failure [15]. Therefore, we argue that an error management strategy mediates the effect of an employee’s daily affective states at work on their learning from failure.

Moreover, cognitive behavior theory suggests that as a boundary condition, an individual’s cognition can have an impact on emotions, affective states, and behavior [16]. Specifically, an individual’s positive cognition of an event will interact with their emotions or affective states to have an impact on their thinking and beliefs after experiencing the event. Previous literature has mentioned that although an individual’s general daily affective states can have an impact on learning behavior, an individual’s perception of work (e.g., the perceived importance of work, organizational identity, etc.) can modify the effects of emotions [17,18]. As a positive cognition toward the specific project, project commitment, in general, refers to “a person’s attachment to a project goal or determination to achieve it” [19] and is a measure of how hard an individual works towards a set goal, and individuals with high levels of project commitment are not willing to easily compromise or abandon their goals, even when faced with hardship and setbacks [19]. Project commitment has often been cited as an important factor influencing employee performance by providing motivation to overcome adversity and achieve goals that seem impossible or far exceed expectations [20]. We believe that project commitment can moderate the relationship between daily affective states and learning from failure by enhancing employees’ sense of responsibility and belonging.

In summary, based on cognitive behavioral theory, we construct a theoretical framework, as shown in Figure 1, to explore how employees’ general daily affective states affect learning from failure via the mediating role of error management strategy and the moderating role of project commitment under the condition of failure (as shown in Figure 1). Our research has made contributions to existing literature in three ways. Firstly, we expand the application of cognitive behavioral theory in the field of organization research, providing important theoretical perspectives for organizational scholars to discuss the impact of the interaction between emotions and cognition on individual behaviors. Secondly, we extend the research on the antecedents of learning from failure, creatively exploring the impact mechanism of the interaction between emotions and cognition on learning from failure. Finally, we creatively explore the important mediated mechanisms of error management strategies, shifting the previous research on error management strategies from the organization level to the individual level, enriching the relevant research on individuals’ failure management strategies.

## 2. Literature Review and Hypotheses

### 2.1. Affective States in Organizational Research

Psychologists generally agree that mood and emotion are important aspects in influencing human behavior patterns [21]. Affective states have a strong influence on social relationships, work attitudes, job performance, and individuals’ memory and learning ability [22,23]. Past research has used the “Big Two” model to classify employee affective states at work into two separate dimensions: positive and negative affects [24].

Broadly speaking, a positive affective state is a general daily affective state accompanied by pleasant emotions in the workplace because daily activities meet personal needs [25], and positive affect influences one’s cognition, such as promoting creativity and accelerating thinking [8]. Positive affective states include feeling determined, focused, positive, inspired, alert, excited, enthusiastic, etc. Previous studies have also confirmed that employees with more stable positive affective states show higher levels of creativity compared to those experiencing neutral states [26]. Tugade and Fredrickson [27] explored the role of positive affective states when employees experience failures and setbacks and concluded that positive affective states improve employees’ resilience to events such as setbacks and help them recover from stress and negative affect; they help them recover quickly and effectively from stressful and negative affective states and adapt to drastic changes in the environment. In contrast to positive affective states, negative affective states are general daily affective states that are not conducive to sustained work or normal thinking by employees under the influence of external or internal factors in a given behavior [28]. The different levels of positive and negative emotions are largely independent of each other. Negative affective states include anxiety, sadness, pain, sorrow, hatred, fear, anger, tension, etc. The literature on cognitive psychology suggests that negative affective states (e.g., anxiety) can impair an individual’s cognitive processing efficiency and executive functioning at the cognitive level [26].

Almost all relevant studies have clearly demonstrated the impact of affective states on learning outcomes, and D’Mello and Graesser [10] emphasized the role of both positive and negative affective states on the involvement in learning activities and underscored how affective states can sustain or disrupt learners’ engagement to impact learning performance. In terms of the “Big Two” affective dimensions, Derakshan [29] argued that negative affective states act as a cognitive load that forcibly occupies the cognitive resources needed for learning, thus affecting the speed of recovery and the effectiveness of learning and memory in the event of frustration, while positive affective states create a sense of security in the environment and lead to more heuristic cognitive processing patterns that promote innovative cognitive formation and learning activities [30]. However, learning from failure, as a specific category of learning, cannot be simply summarized as “learning cognition in a state of frustration”. Most of the attention is focused on the organizational level on the effect of affective states on learning from failure, while there is a lack of individual and team level research, and no scholars have yet studied the impact of general daily affective states on learning from failure as a specific category at the individual level. Therefore, based on existing research, we posit that there may also be different effects of employees’ daily affective states on learning from failure.

In addition, cognitive behavioral theory emphasizes the role of cognitive activity in psychological or behavioral problems, which suggests that cognition plays an important role of a “coordinator” between affective states and individual behaviors [16]. In particular, positive cognition helps individuals to perceive the effects of affective states and to take positive behaviors, while negative cognition impedes individuals’ ability to perceive the effects of affective states and to take negative behaviors [31]. We notice that in existing studies, project commitment, as an individual’s positive cognition toward a project, has not received much attention from scholars studying failure. Most studies have explored the direct impact of project commitment on individuals’ positive behavior [32,33], while neglecting its crucial positive boundary role in the bond of “emotional affects—behaviors”. Therefore, we intend to include project commitment in our theoretical model to explore how this positive cognition interacts with daily affective states to have an impact on the process of affective states influencing learning from failure.

### 2.2. Positive Affective States and Error Management Strategy

As error management emphasizes positive responses to errors, error management strategies are the most efficient means to deal with errors in a changing and complex work environment [34]. Positive affective states have positive effects on individuals’ cognition and thinking, such as by helping individuals overcome negative states after failures faster, helping individuals to develop their thinking and maintain stability in their mental states, and helping individuals more effectively accumulate resources [30]. Therefore, we believe that positive affective states help individuals better face, analyze, and deal with errors and have a higher tolerance for possible future errors, i.e., by adopting an error management strategy.

First, positive affective states can help individuals maintain strong interpersonal relationships and drive them to communicate openly about errors. Positive affective states encourage individuals to be open with the outside world, help individuals maintain good interpersonal relationship after a failure, and bring linking resources to individuals to enhance social connections. Individuals are then motivated to engage in communication about errors [35] and receive more information about errors. At the same time, positive affective states expand an individual’s cognitive and action range; broaden an individual’s instantaneous perception, thinking, and activity sequences [8]; and enable the rapid integration of this obtained information, thus helping an individual process errors faster [36], and ultimately, an individual becomes more effective at error management.

Second, positive emotions can help employees maintain flexible thinking and a positive mindset and pursue external exploration. After a failure occurs, an individual’s exploration and experimentation in response to an error may increase as he or she tries to resolve the failure [37], but these exploratory activities may not last long because failure itself is a form of negative feedback that may cause an individual to feel frustrated [38]. Positive affective states and cognitive evaluations help people recover quickly from negative affective states [27] and maintain a positive mental state and vitality to continue their external exploration. Therefore, through exploration and experimentation, individuals may eventually establish a means to effectively cope with errors, buffering negative emotions psychologically and physiologically, helping individuals accept the results of failure and find more solutions, and ultimately reducing the negative effects of errors after they occur.

In summary, positive affective states have positive impact on all three modes of error management strategy implementation, i.e., they help individuals analyze, communicate, and explore more solutions to errors and effectively learn from them. That is, under the influence of positive affective states, employees tend to adopt error management strategies. Therefore, we propose the following hypothesis.

**H1a:** *Positive affective states are positively related to error management strategy*.

### 2.3. Negative Affective States and Error Management Strategy

Negative affective states will place an individual in a long-term negative state of self-denial and self-doubt, which is not conducive to subsequent psychological and physiological adjustment [39]. Therefore, we argue that negative affective states will prevent an individual from adopting an error management strategy in the face of errors.

First, negative affective states occupy individual emotional resources, making it difficult for individuals to dedicate resources to managing errors. Modern stress theory states that negative affective states will place additional burdens and tasks on individuals, causing the information processing resources allocated by individuals to other thinking activities to become overburdened [40]. At the same time, individuals in negative affective states are more inclined to shift all attention resources to their own affective exhaustion than to reflect and learn from errors [41,42]. This will interfere with employees’ effective handling of errors after errors occur, magnify the negative consequences of errors, and affect the development of error management strategies.

Finally, negative affective states affect communication between individuals and the outside world, thus hindering learning. Negative affective states can cause individual expression inhibition and reduce feedback and communication in interpersonal communication [43]. Being in negative affective states will increase the emotional sensitivity of individuals to conflict and have a negative impact on the communication objects [44], which is not conducive to normal communication, causing individuals to receive more negative feedback in communication. Such a cycle hinders the acquisition of information and error management.

Negative affective states have a negative impact on individuals’ perceptions, resource allocation, and external interactions and hinder the advancement of error management; therefore, this paper proposes the following:

**H1b:** *Negative affective states are negatively related to error management strategy*.

### 2.4. Error Management Strategy and Learning from Failure

The basic assumptions of error management emphasize that error management can help employees grow from their mistakes; an efficient error management strategy aims to help individuals in organizations change rather than be stagnant due to intimidation [45]. According to previous studies, there are three main ways to implement error management strategy: (1) effectively identifying and analyzing errors and communicating openly about them; (2) effectively handling errors and reducing the negative consequences of errors; and (3) learning about errors and viewing them as valuable learning opportunities [46]. Thus, we argue that error management will provide opportunities for learning from failure and help individuals pursue new directions and develop exploratory and creative solutions from their failures to help them learn from failure.

First, the implementation of error management strategy can improve individuals’ tolerance of errors, advance the learning process, and facilitate learning from failure. Within the error management system, the way errors are handled is more important to error management than handling the errors themselves [47]. Since errors are inevitable for anyone in any environment, inevitable “strategic failures” need to be addressed, and a foundation for learning from failure needs to be built [48]. Error management strategy involves employees shifting their focus from the negative effects of failure to the positive value behind it, creating a space for employees to make mistakes [49]. Additionally, error management strategy facilitates individuals tolerating failure. In conclusion, error management strategies improve individual tolerance of errors and advance the learning process, thus contributing to the development and improvement of learning from failure.

Second, EMS implementation can foster positive error response thinking following a failure, improve individual learning, and stimulate long-term learning from failure processes. The positive effects of error management are mainly reflected in the development of positive error response thinking, which stimulates a learning orientation towards goals [46] and helps employees overcome inherent stereotypical thinking and move from old mechanical practices to new directions. With the help of this type of thinking, exploratory, creative solutions have a better environment for emergence [37]. In terms of individual learning ability, an important catalyst of new knowledge acquisition and learning ability improvement is failure and alternative experiences [50]. With a strong error management strategy, employees not only actively think about and analyze their own error experiences, but also actively communicate and share with other employees in the organization. In communicating about errors to benefit from valuable successful experiences of others, they can not only develop their own error response thinking, but also acquire knowledge and then learn from failure.

Finally, the implementation of error management strategy will help failures be transformed into effective learning opportunities. Adopting an error management strategy can bring about a high level of initiative and willingness to explore [12], and it focuses on the positive effects of errors on individuals. Individuals enhance the spirit of exploration through the ability of dealing with errors and thinking of errors, which in turn promotes learning from failure and an understanding of means of production [46]. Therefore, error management helps those who have experienced failure find new learning opportunities and ideas and promotes learning from failure.

In summary, we propose the following:

**H2:** *An error management strategy is positively related to learning from failure*.

**H3a:** *Employees’ positive affective states are positively and indirectly related to learning from failure, *via* the mediating effect of error management strategy*.

**H3b:** *Employees’ negative affective states are negatively and indirectly related to learning from failure, *via* the mediating effect of error management strategy*.

### 2.5. Error Management Strategy, Project Commitment, and Learning from Failure

Previous research has argued that project commitment can enhance employees’ identification with the organization, thereby engaging them in active learning and other positive behaviors to achieve project goals [51]. Thus, we contend that people who possess a strong project commitment can alter their outlook on failure, utilizing positive actions and cognitive modifications, enhancing the positive impact of positive affective states and weakening the negative impact of negative affective states.

First, project commitment can enhance employees’ sense of responsibility to a project, rendering individuals willing to devote more energy to thinking about errors, exploring solutions, etc. [20]. Employees high in project commitment are able to persevere in the face of setbacks or adversity without giving up or compromising project goals and continue to work until such goals are achieved [19,20]. They may promote the achievement of project goals through communicating or sharing knowledge with their colleagues [52]. Therefore, employees with project commitments can quickly obtain information from failed events, which can help them adjust behaviors in a timely manner and reduce adverse effects.

Moreover, employees with a high sense of project commitment will pay more attention to organizational goals and behavioral norms. In order to achieve goals, they will transform hard work (or applying effort to complete a task) into an obligatory behavior, and this internalized behavioral norm can prompt employees to consciously change their behavior patterns to meet team goals and performance requirements [53]. Driven by this belief, employees with high project commitments will adjust their behavior in a timely manner to achieve better performance and reduction [33].

Third, in instances of failure, employees high in project commitment show a strong sense of responsibility to a project that differs from that of the average employee [54], and in this case, they tend to devote more resources and energy to error management. They derive pleasure from activities such as knowledge-sharing for the organization and actively engaging in team-related endeavors [55], which can enhance the relationship between positive affective states and error management strategy. On the contrary, project commitment can weaken the negative relationship between negative affective states and error management strategy.

In conclusion, project commitment will moderate the relationship between affective states (both positive and negative) and learning from failure. Specifically, for individuals with higher levels of project commitment, the positive effect of positive affective states on learning from failure will be stronger, while the negative effect of negative affective states on learning from failure will be mitigated.

**H4a:** *Project commitment moderates the relationship between positive affective states and EMS. When the level of project commitment is high, the positive relationship between positive affective states and error management strategy will be enhanced*.

**H4b:** *Project commitment moderates the relationship between negative affective states and EMS. When the level of project commitment is high, the negative relationship between negative affective states and error management strategy will be weakened*.

Employees high in project commitment have a greater sense of responsibility to a project, and they tend to devote more resources and energy to error management when influenced by their emotions [53]. At the same time, employees high in project commitment emphasize trust in their teams and a sense of responsibility among team members, resulting in stronger interpersonal communication and helping employees better utilize their positive emotions in error management [34]. This also helps employees overcome the external communication problems that negative affective states create in error management and learning from failure and then weakens the negative effect of negative affective states from the start. Thus, the mediating role of error management strategy in positive affective states affecting learning from failure is strengthened in this process, while the mediating role of negative affective states affecting learning from failure is weakened.

Therefore, we conclude with the following hypotheses.

**H5a:** *The negative and indirect effect of negative affective states on learning from failure *via* EMS is moderated by project commitment, such that the negative indirect effect is stronger when project commitment is low and is weaker when project commitment is high*.

**H5b:** *The positive and indirect effect of positive affective states on learning from failure *via* EMS is moderated by project commitment, such that the positive indirect effect is stronger when project commitment is high and is weaker when project commitment is low*.

## 3. Research Design

### 3.1. Participants and Data Collection

Since most high-tech industries create creative work “out of nothing”, they are more likely to face failure, and learning from failure is more important for them. Therefore, this study considers the project members of high-tech R&D companies (those with annual sales of high-tech products (services) accounting for 60% of their total sales in the past year and with at least 10% of their R&D staff coming from the company) as the target population. From mid to late 2020, we randomly selected 400 companies from the list of high-tech companies reported in Beijing and invited these companies to participate in the study by telephone. In communicating with the CEOs of these companies, we explained the purpose of the study, emphasized the confidentiality of the data collected, and promised to provide feedback to the company leaders on the results of the study. For the companies that eventually confirmed their participation in the survey, with the help of an internal coordinator appointed by the CEO, the research assistant first identified the participating high-tech team members and then distributed the questionnaires and asked for their help in filling them out before holding regular meetings with these members. For those who were absent, the research assistant obtained their contact information from the internal coordinator and ensured the return of the questionnaires through follow-up communication. In addition, to increase the questionnaire response rate, the CEO of each company was asked to provide a letter of endorsement in support of the study, and a small gift was distributed along with the questionnaire.

In the end, 58 companies participated in the final survey, yielding a response rate of 14.5%. We used an independent sample *t*-test to compare variables, such as the type of companies, between the participating and non-participating companies, and there was no significant difference between the two (see Table A2 for specific results). A final sample of 776 participants completed the survey. Due to missing data, the final valid sample included 774 participants. The average age of the participants was 31.65 years old, approximately 77% of them were male, more than 92.4% of the respondents had a bachelor’s degree or higher, and more than 41.1% had a master’s degree or higher. Since the data were collected after the outbreak of the COVID-19 pandemic, there were no significant organizational or contextual changes in all participating organizations, which means that our data collection time will not have a significant impact on the research.

### 3.2. Measurement

In this study, established scales were selected to measure the study variables, and the questionnaire was translated using the back translation method (i.e., all questions were first translated from English to Chinese and then back-translated into English by two Ph.D. students independently) to ensure the accuracy of the translation [56]. In addition, we define project failure as the termination of a project because the project failed to meet its objectives [37], and this definition was provided in the introduction section at the beginning of the questionnaire to ensure that participants had a better understanding of the purpose of the study. The following variables were measured in this paper.

#### 3.2.1. Positive and Negative Affective States

In this study, we use the PANAS scale developed by Thompson [57] and the PID scale developed by Mackinnon et al. [58] to measure employees’ general affective states in their daily work. Positive affective state variables measured include the following: alert, inspired, determined, attentive, and active (the PANAS scale) and excited and enthusiastic (the PID scale). The Cronbach α of the scale was measured as 0.796. Negative affective state variables measured include the following: upset, hostile, ashamed, nervous, and afraid (the PANAS scale) and scared and distressed (the PID scale). Options range from 1 (completely disagree) to 6 (completely agree). The Cronbach’s α of the scale was measured as 0.782.

#### 3.2.2. Error Management Strategy (EMS)

According to Van Dyck et al. [12], there are two main types of error coping behavioral approaches: error management strategy and error aversion strategy, both of which have different emphases. In this study, the error coping scale developed by Van Dyck et al. [12] was used to measure error management strategy and error aversion strategy. The error management strategy was measured with 17 items. Representative items include the following: “After an error, I think through how to correct it” and “After making a mistake, I try to analyze what caused it”. The error aversion strategy was measured with 11 items. A representative item is “There is no point in discussing errors with my colleagues”. Options range from 1 (completely disagree) to 6 (completely agree). The Cronbach’s α of the scale was measured as 0.885.

#### 3.2.3. Project Commitment

In this study, the five-item scale developed by Hoegl et al. [20] was used to measure project commitment. Representative items include “I feel fully responsible for achieving the common project goals”, “I committed not only to my teams, but to the overall project”, and so on. Options range from 1 (completely disagree) to 6 (completely agree). The Cronbach’s α of the scale was measured as 0.861.

#### 3.2.4. Learning from Failure

In this study, the eight-item scale developed by Shepherd et al. [37] was used to measure learning from failure. Statements measured include “I can more effectively run a new project” and “I can ‘see’ earlier the signs that a project is in trouble”. Options range from 1 (completely disagree) to 6 (completely agree). The Cronbach’s α of the scale was measured as 0.904.

#### 3.2.5. Control Variables

To avoid the impact of individual differences, we controlled for demographic variables, including gender, age, and education level. In addition, we controlled for the tenure of individuals in the company and their understanding of the reasons for failure. These individual work-related variables are closely associated with failed projects and may influence individuals’ processing of emotions and corresponding behavioral responses [59].

### 3.3. Confirmatory Factor Analysis

To test the model fit, we conduct a series of confirmatory factor analyses (CFAs) of the six variables of positive affective states, negative affective states, error management strategy, error aversion strategy, project commitment, and learning from failure to test the discriminant validity of the six core variables, and the results are shown in Table 1. As shown, the theoretical model (6-factor model) presented in this paper had the best data for each fit indicator relative to several other models (CMIN/DF = 2.26, IFI = 0.91, TLI = 0.90, CFI = 0.91, and RMSEA = 0.04). This indicates that the model proposed in this study is valid.

### 3.4. Testing Validity

We further calculated the construct reliability (CR) of each scale. We used the average variance extracted (AVE) to assess convergent validity to ensure a good level of reliability and validity. As shown in Table A1, the CR of all measured variables exceeds 0.70, and the AVE exceeds the recommended value of 0.50, denoting the strong convergent validity of the scale. In addition, we used the Heterotrait–Monotrait Ratio (HTMT) to calculate the discriminant validity of the model. As shown in Table 2, the HTMT values between each study variable are all less than 0.85, indicating the good discriminant validity of the theoretical model [60].

## 4. Hypothesis Testing

We performed a hierarchical regression analysis using SPSS 26.0 for hypothesis testing. The data obtained for the study were collected concurrently, so there may be a risk of common method bias (CMB). To address the issue of CMB due to our single round of data collection, we conducted Harman’s one-way test [61] and found 4 factors with characteristic roots of greater than 1 and that the first factor had an explanatory rate <40%. Thus, it can be concluded to some extent that no serious problem of homogeneous method bias affects this study.

We used SPSS 26.0 and AMOS 24.0 as tools for data analysis and hypothesis testing to analyze the basic data distribution and correlations, to perform reliability testing of the scales, and to conduct a multiple linear regression analysis. The results of the study are as follows.

### 4.1. Descriptive Statistics and Correlation Coefficient Test

We used SPSS 26.0 statistical software to perform descriptive statistical tests for each variable, including the calculation of means and standard deviations, and then create a correlation coefficient matrix, and the results are shown in Table 3. Table 3, which provides descriptive statistics and correlation coefficients, includes means, standard deviations, alpha coefficients, and correlations between the study variables. From the table, we find that positive affective states are positively correlated with learning from failure (*r* = 0.19, *p* < 0.01), and negative affective states are negatively correlated with learning from failure (*r* = −0.16, *p* < 0.01). Positive affective states are positively associated with EMS (*r* = 0.28, *p* < 0.01), and negative affective states are negatively associated with EMS (*r* = −0.19, *p* < 0.01). EMS is positively associated with learning from failure (*r* = 0.42, *p* < 0.01); thus, our proposed hypothesis is initially supported.

### 4.2. Testing Mediating Effects

We conducted a multiple linear regression analysis with SPSS 26.0 to examine the hypotheses, and the results are shown in Table 4. As shown in Model 2.2, the direct effects of employees’ positive/negative affective states and learning from failure are both significant. Positive affective states have a positive relationship with learning from failure, while negative affective states have a negative relationship with learning from failure (*b* = 0.17, *p* < 0.001; *b* = −0.12, *p* < 0.01). As shown in Model 1.2, positive affective states have a positive relationship with EMS (*b* = 0.26, *p* < 0.001); negative affective states have a negative relationship with EMS (*b* = −0.15, *p* < 0.001).

As shown in Model 2.3, EMS has a positive relationship with learning from failure (*b* = 0.38, *p* < 0.001). In addition, positive affective states were insignificantly positively correlated with learning from failure (*b* = 0.05, *p* > 0.05), negative affective states were insignificantly negatively correlated with learning from failure (*b* = −0.04, *p* > 0.05), and the correlation coefficient was reduced, indicating that EMS played a fully mediating role.

We used the bootstrap method to estimate this indirect effect. The macro program PROCESS developed by Hayes was used to perform the bootstrap test. On the basis of the original data (*N* = 774), 1000 bootstrap samples were randomly selected by repeated random sampling, and the 95% confidence interval was obtained from the estimated value of the mediation effect. When we apply positive affective states as the independent variable, the estimated value of the indirect effect is 0.14, the mediating effect accounted for 64.5% of the total variance, and the bias-corrected 95% CI of the estimated effect does not include zero [0.09, 0.19], indicating that error management partially mediates this positive mediated relationship. When we apply negative affective states as the independent variable, the estimated value of the indirect effect is −0.08, the mediating effect accounted for 55.4% of the total variance, and the bias-corrected 95% CI of the estimated effect does not include zero [−0.13, −0.04], indicating that error management partially mediates this negative mediated relationship. Therefore, Hypotheses 1a, 1b, 2, 3a, and 3b are supported.

### 4.3. Testing Moderating Effects

We conducted an interaction effect test to test Hypotheses 4a and 4b. As shown in Table 3, when the moderating variables were included in the model, there was no significant positive correlation between project commitment × positive affective states and EMS (*b* = −0.04, *p* > 0.05), and there was a significant positive relationship between project commitment × negative affective states and EMS (*b* = 0.07, *p* < 0.05). This indicates that project commitment has no significant moderating effect on positive affective states influencing EMS, while project commitment has a significant moderating effect on negative affective states influencing EMS.

We then conducted a moderated mediation effect test using the macro program PROCESS. When using negative affective states as the independent variable, the index value of the moderated mediating effect is 0.0524, and the bias-corrected 95% CI of the estimated effect does not include zero [0.00, 0.11]. It can be inferred that there is a significant moderating mediating effect in the model. When using positive affective states as the independent variable, the bias-corrected 95% CI of the estimated effect does include zero [−0.08, 0.04], indicating that the moderating variable project commitment does not have a significant moderating effect in the model. To better interpret the moderating effect of project commitment, we defined employees’ high and low levels of project commitment as one standard deviation above and below the mean, respectively. As shown in Figure 2, for employees lower in project commitment, negative affective states significantly and negatively influenced EMS (simple slope = −0.20, *t* = −4.51, *p* < 0.001); for employees higher in project commitment, the negative predictive effect of negative affective states on EMS was diminished (simple slope = −0.05, *t* = −1.07, *p* > 0.05). Therefore, Hypothesis 4b is supported, while Hypothesis 4a is not supported. Hypotheses 5a and 5b propose the moderated mediation model, with the mediation process of affective states and learning from failure through an EMS moderated by project commitment. The index of moderated mediation (negative affective states) is significant (Index = 0.03, BootLLCI = 0.004, BootULCI = 0.057), while the index of moderated mediation (positive affective states) is insignificant (Index = −0.02, BootLLCI = −0.083, BootULCI = 0.038). Therefore, Hypothesis 5b is supported, while Hypothesis 5a is not supported.

Because of the insignificant moderating effect of high-project-commitment subgroups, we also plotted the Johnson–Neyman (J-N) slope plot (Figure 3) to explore the significant domain of project commitment as a moderating variable, and the results show a significant moderating effect of the project commitment in the relationship between negative affective states and EMS when, and only when, project commitment takes a value of less than 4.98. However, whether significant or not, the negative predictive effect of negative affective states on EMS diminishes with increasing levels of project commitment in the full domain. Thus, taken together, the effect of negative affective states on learning from failure through EMS strategy is moderated by project commitment.

## 5. Discussion

We combine previous research to build a theoretical framework and empirically analyze questionnaire data drawn from project teams in high-tech enterprises to draw the following conclusions. First, positive and negative affective states positively and negatively influence the effect of learning from failure, respectively. At the same time, we found that the error management strategy plays a mediating role in the process of affective states affecting learning from failure, i.e., positive and negative affective states positively and negatively affect the error management strategy, respectively, while the use of an error management strategy positively affects learning from failure. In addition, we identify a moderating role of project commitment in the mechanism by which affective states influence error management strategy. On the one hand, a moderating effect of project commitment on the relationship between positive affective states and error management strategy was not confirmed. On the other hand, the moderating effect of project commitment on the relationship between negative affective states and error management strategy is supported; that is, it weakens the negative effect of negative affective states on error management strategy.

### 5.1. Theoretical Contributions

First, we extend the research on the antecedents of learning from failure and identify the roles of employees’ daily affective states in learning from failure. Although some scholars have focused on the important role of emotions (e.g., anger [62]; shame [63]) in the impact of learning from failure, these research perspectives are relatively single. Most of them focus on a single negative dimension of emotional characteristics, lacking paying attention to the significant role of positive emotions and affective states. For example, Shepherd et al. [37] explored the influence of negative affective states on the effect of learning from failure experience. Moreover, Shepherd, Covin, and Kuratko [41] discussed, from the perspective of social cognition, how the effect of negative affective states (e.g., anxiety) negatively affects the individual’s learning from the failure process. The effects of positive affective states are selectively ignored. Differently from previous research, we focus on general daily affective states rather than the affective states triggered by failure. In this paper, we integrate the two dimensions of affective states and behaviors and identify different mechanisms of affective impact on learning from failure, creatively open up the theoretical path from “affective states to learning from failure”, and enrich the discussion on the impact of positive emotions and affective states on learning from failure processes.

Secondly, we further enrich relevant research on error coping strategies. In fact, most of the literature has examined error management strategy on an organizational level and tested the consequences on organizations (e.g., organizational performance [12]), teams (e.g., team creativity [64]), and employees (e.g., employee innovation and performance [65]). Recently, although some scholars have explored the boundary role of error management culture on individuals’ learning from failure [66], limited research has been conducted to identify the factors influencing the implementation of error management strategy at the individual level through the correspondence of the three antecedent pathways of error management strategy. We combined previous research with the introduction of error management strategy to create a clear pathway model of the influence of positive and negative affective states on individuals’ learning from failure, validated the hypotheses in the model based on the data, and gave the data results to analyze the influence of affective states on error management strategy. We creatively shift the perspective of error management culture from an organizational level to the individual level, providing a new perspective on the study of error management strategy.

Finally, by innovatively introducing cognitive behavior theory into the field of organizational management, we explored how project commitment, as an individual’s positive cognition toward project work, can influence learning from failure by interacting with individuals’ daily affective states. Cognitive behavioral theory is currently mostly used in the field of psychology to explore the effects of individuals’ cognition on emotions and behavior. Based on this, many psychologists have used cognitive behavioral therapy to address individual psychological barriers [67,68]. However, organizational scholars have not yet applied cognitive behavior theory in the research to address the interaction between cognition and emotions in employees’ work behavior. Therefore, we introduced cognitive behavior theory into this study, expanding its application in the fields of organizational management and organizational psychology. Notably, although some scholars have explored how project commitment can bring various positive behaviors to employees (such as knowledge sharing [34], team collaboration [20], and team performance [69]), how project commitment interacts with affective states as a boundary condition has not yet received attention from scholars. It is with reference to previous research on project commitment that this paper proceeds, linking affective states and behaviors through project commitment, pioneering the use of project commitment as a boundary condition for affective states affecting error management strategy. This can inform subsequent research on project commitment.

However, in our study, project commitment was only able to moderate the effect of negative affective states on error management strategy. We attribute this to the fact that when individuals display positive attitudes or behaviors, it often means that the resources provided by the job meet the individual’s needs [70]. Project commitment at this point does not enable the employee to invest further resources into the job. Thus, there is no significant moderating effect of project commitment in the effect of positive affective states on error management strategy. Whether project commitment can play a general mediating or moderating role between affective states and individual behavior needs to be demonstrated in future research.

### 5.2. Practical Implications

The results of this paper are of great practical significance for managers and organizations. First, our results show that affective states have a significant impact on employees’ learning from failure. When failure occurs, managers should strengthen the emotional monitoring of their organizations as a whole. At the same time, the soft qualities of employees during emotional work events can also be included in the scope of investigations of employees. Second, the use of an error management strategy will significantly improve the failure learning effect of employees at the individual level. Actively carrying out error management in training is a new means to promote employees’ learning from failure that pays attention to the positive effect of errors on individuals. Finally, enterprises should cultivate employees’ project commitment. This study shows that different levels of project commitment have different effects on employees in different affective states. Enterprises should adjust the project commitment levels of employees in a timely manner by adjusting resource investment and cultivating employees’ sense of belonging according to the given situation. Finally, we would like to say that failure knows no borders. Although our participants come from Chinese organizations, with the increasing economic and commercial uncertainty after COVID-19, failure has become an opportunity and challenge faced by various countries, societies, organizations, and individuals. Therefore, all organizations and individuals should attach importance to learning from failures and managing failures, in order to achieve long-term success.

### 5.3. Limitations and Future Research Directions

This paper draws conclusions from an empirical study of questionnaires, but it still has several limitations. First, employees’ affective states are dynamic and change over time with a halo effect, and the cross-sectional questionnaire used in this study makes it difficult to accurately measure the changes in affective states that occur when employees actively adjust. Therefore, future studies can adopt experience sampling methods, experimental methods, or in-depth interview methods to further verify the findings of this paper and explore causal relationships and their influencing mechanisms. Second, we neglect comprehensiveness of the mediating effects and boundary effect, and only using error management strategy as a mediating variable and project commitment as a moderating variable. Whether there are other mediating and boundary effects among these effects is not determined in this study, and subsequent studies can focus on the exploration of other potential influencing factors. Finally, the measurement instruments used in this study are based on established foreign scales. Although the questionnaire has passed our reliability test, we do not know whether the instrument truly reflects the organizational cultural differences, the professional specificities of the researchers, and the cultural differences between eastern and western countries. Therefore, future research can further explore and improve the measurement tools used in this paper to develop specific measurement tools that are both scientific and innovative.

## Figures and Tables

**Figure 1 behavsci-13-00514-f001:**
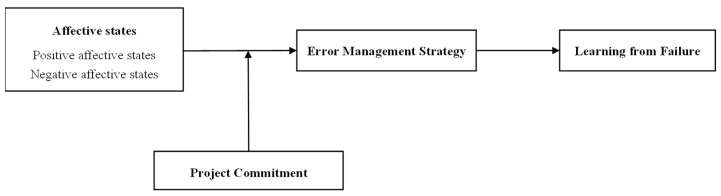
Theoretical framework diagram.

**Figure 2 behavsci-13-00514-f002:**
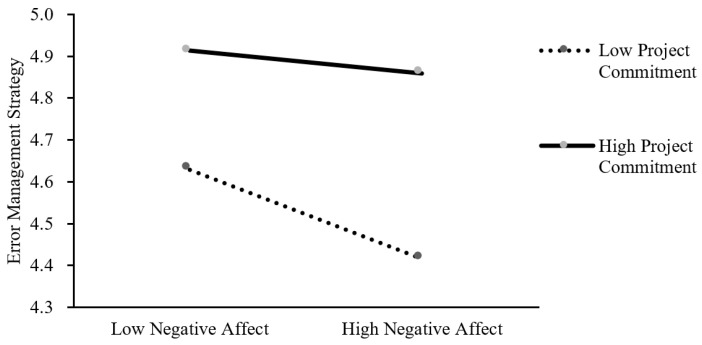
The moderating role of project commitment in the relationship between negative affective states and error management strategy.

**Figure 3 behavsci-13-00514-f003:**
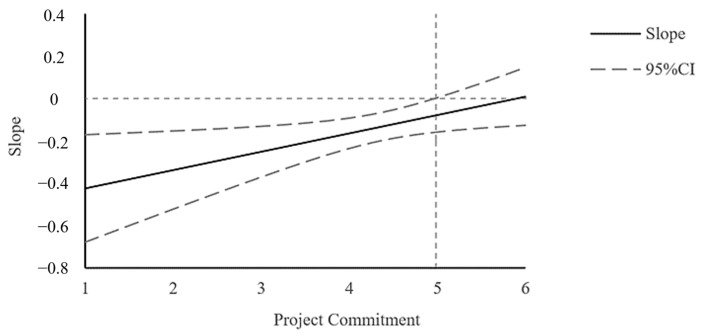
J-N slope diagram for the moderating effect of project commitment.

**Table 1 behavsci-13-00514-t001:** Confirmatory factor analysis (CFA).

Models	CMIN	DF	CMIN/DF	IFI	TLI	CFI	RMSEA
Six-factor model (PA, NA, EMS, EAS, PC, LFF)	3144.86	1391	2.26	0.91	0.90	0.91	0.04
Five-factor model 1 (PA + NA, EMS, EAS, PC, LFF)	4101.66	1396	2.94	0.86	0.85	0.86	0.05
Five-factor model 2 (PA, NA, EMS + EAS, PC, LFF)	4972.57	1396	3.56	0.82	0.80	0.81	0.06
Four-factor model (PA + NA, EMS + EAS, PC, LFF)	5930.65	1400	4.24	0.77	0.75	0.76	0.07
Three-factor model (PA + NA + PC, EMS + EAS, LFF)	7233.90	1403	5.16	0.70	0.68	0.70	0.07
Two-factor model (PA + NA + PC, EMS + EAS + LFF)	8608.21	1405	6.13	0.63	0.60	0.63	0.08
One-factor model (PA + NA + PC + EMS + EAS + LFF)	9356.29	1406	6.66	0.59	0.56	0.59	0.09

Note: PA indicates positive affective states; NA indicates negative affective states; PC indicates project commitment; EMS indicates error management strategy; EAS indicates error aversion strategy; LFF indicates learning from failure. Sample size: 774.

**Table 2 behavsci-13-00514-t002:** HTMT values between study variables.

Variables	1	2	3	4	5
1 Positive affective states					
2 Negative affective states	0.38				
3 Error management strategy	0.38	0.24			
4 Error aversion strategy	0.36	0.32	0.33		
5 Project commitment	0.31	0.16	0.50	0.25	
6 Learning from failure	0.28	0.20	0.48	0.25	0.74

Note: The number in front of each variable and the first line represents the sequence number of the variable.

**Table 3 behavsci-13-00514-t003:** Descriptive statistics and correlation coefficient matrix.

	Means	SD	1	2	3	4	5	6	7	8	9	10	11
1 Gender	0.77	0.42											
2 Age	31.65	5.50	0.04										
3 Education Level	4.36	0.68	0.00	0.19 **									
4 Work Tenure	1.76	0.73	0.01	0.69 **	0.19 **								
5 Understanding of the Reasons of Failure	2.93	0.96	0.01	−0.05	−0.10 **	−0.08 *	(0.870)						
6 Positive Affective States	3.58	0.57	−0.03	0.03	−0.05	−0.02	0.01	(0.796)					
7 Negative Affective States	2.09	0.53	0.05	0.05	0.06	0.12 **	0.06	−0.17 **	(0.782)				
8 Error Management Strategy	4.70	0.53	−0.02	0.02	0.05	0.01	−0.03	0.28 **	−0.19 **	(0.885)			
9 Error Aversion Strategy	2.97	0.77	−0.01	0.03	0.02	0.05	0.06	−0.22 **	0.26 **	−0.14 **	(0.852)		
10 Project Commitment	4.44	0.86	−0.02	0.01	−0.07 *	−0.01	0.01	0.23 **	−0.09 *	0.39 **	−0.17 **	(0.863)	
11 Learning from Failure	4.58	0.84	−0.05	−0.02	−0.04	−0.03	−0.07 *	0.19 **	−0.16 **	0.42 **	−0.18 **	0.48 **	(0.909)

Note: * *p* < 0.05; ** *p* < 0.01. Sample size: 774. The number in front of each variable and the first line represents the sequence number of the variable.

**Table 4 behavsci-13-00514-t004:** Regression analysis results.

Variables	Error Management Strategy				Learning from Failure		
	Model 1.1	Model 1.2	Model 1.3	Model 1.4	Model 2.1	Model 2.2	Model 2.3
1 Gender	−0.06	−0.02	−0.01	0.00	−0.13	−0.10	−0.10
2 Age	0.18	−0.02	−0.03	0.03	0.10	−0.04	−0.04
3 Education Level	0.06	0.10	0.13 **	0.13 **	−0.07	−0.04	−0.08
4 Work Tenure	−0.03	0.03	0.02	0.00	−0.05	−0.01	−0.02
5 Understanding of the Reasons of Failure	−0.03	−0.02	−0.02	−0.02	−0.08 *	−0.07 *	−0.06
6 Positive Affective States		0.26 ***	0.18 ***	0.17 ***		0.17 ***	0.05
7 Negative Affective States		−0.15 ***	−0.13 ***	−0.13 ***		−0.12 **	−0.04
8 Project Commitment			0.34 ***	0.35 ***			
9 Project Commitment × Positive Affective States				−0.02			
10 Project Commitment × Negative Affective States				0.07 *			
11 Error Management Strategy							0.38 ***
12 Error Aversion Strategy							−0.11 **
*R* ^2^	0.00	0.10	0.20	0.21	0.01	0.05	0.19
△*R*^2^		0.10	0.10	0.11		0.05	0.14
*F*	0.62	12.56	25.79	21.32	1.79	7.04	21.45

Note: * *p* < 0.05; ** *p* < 0.01; *** *p* < 0.001; Sample size: 774. The number in front of each variable and the first line represents the sequence number of the variable.

## Data Availability

The datasets generated during and/or analyzed during the current study are available from the corresponding author on reasonable request.

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
