# Peer review of "Are You Dominated by Your Affects? How and When Do Employees’ Daily Affective States Impact Learning from Project Failure?"

_behavsci, 2023, doi:10.3390/bs13060514_

Round 1
Reviewer 1 Report
Dear Authors,
The article is scientifically interesting and has numerous research and empirical values.
The Introduction is too extensive, which means that it does not properly fulfill its role in the article.
It should be simplified. The research problem should clearly result from the introduction. The description of the research at this point should be synthetic, it is too long. The authors should focus more on the presentation of the research question, and the scope of the research, and not on the detailed description of the research.
The hypotheses were formulated correctly and then verified. The results of the research were presented in a proper way.
The tables and figures have been corrected. The discussion is too short, it should be extended. Please refer to the previous research in this field.
Section 5.1. Theoretical contributions have not been properly adjusted. This should be corrected.
In summary, please:
- Rephrase the introduction
- Extend the discussion
- Make editorial corrections in point 5.1.
Author Response
Dear reviewer,
Thank you for your constructive and valuable comments on the article. We have made revisions to all comments one by one, and have updated and improved the article based on the comments, which has greatly improved the quality of our article. The followings are our responses to your comments one by one:
Point 1: The introduction is too extensive, which means that it does not properly fulfill its role in the article. It should be simplified. The research problem should clearly result from the introduction. The description of the research at this point should be synthetic, it is too long. The authors should focus more on the presentation of the research question, and the scope of the research, and not on the detailed description of the research.
Response 1: Thank you for your comment! We have tried to streamline the content of the existing Introduction as much as possible. However, according to the request of another reviewer, we had to add additional contents, such as a brief introduction of the theory and a brief theoretical contribution. Therefore, overall, the length of the introduction has not been significantly reduced. But we indeed have rephrased the introduction, and tried to make the logic clearer. You can refer to the revised manuscript for specific content.
Point 2: The discussion is too short, it should be extended. Please refer to the previous research in this field. Section 5.1. Theoretical contributions have not been properly adjusted. This should be corrected.
Response 2: Thank you for your valuable comment! After reading the theoretical implication again, we found that this section was indeed described very simply, and we did not have a good dialogue between the research results and the literature. Therefore, we have rewritten the section of theoretical significance to form a communication between the research results and existing literature. The specific content is shown in Page 14, Line 582.
Reviewer 2 Report
Thank you so much for inviting me to review manuscript behavsci-2406031 entitled “Are you dominated by your effects? How and when do employees’ daily affective states impact learning from project failure.” I read it with great interest but there are shortcomings. Following are my comments
Abstract
v In the first two lines, please explain the problem.
v Add a line on the analysis technique.
Introduction:
· In general, it is well written. However, I would like to know what is already (if any) available in the literature on the affective states and learning relationship.
· The arguments related to the inclusion of moderating variables need to be improved.
· Furthermore, briefly explain the theoretical implications and novelty of the study.
Literature review:
· There should be one paragraph about Cognitive behavior theory and how it supports the proposed relationships.
· There are many statements/claims without the support of literature. I suggest supporting your claim in the light of literature.
· In addition, use the adopted theory in the development of the proposed hypothesis.
· Furthermore, the arguments for mediating hypotheses are not clear. More specifically, hypothesis 3b cannot be accepted because if it includes a negative value then the confidence interval would have zero. Therefore, it must be rejected.
· Moreover, the explanation of moderating hypotheses is not convincing and required a lot of work.
· Besides, most of the cited articles are old. It should be improved with the help of recently published papers (not more than 5 years old).
Methods, analysis, and Results:
I appreciate the good sample size but I have some queries and suggestions.
Ø When the data were collected?
Ø The scales should be added in the appendix for clarity.
Ø What was the response rate? Did you check non-response bias?
Ø Add a table and show the factor loading of each item, AVE, and CR of each scale.
Ø Correlation is fine but I think HTMT should also be added.
Ø The R-square is very small? Could you please justify this?
Ø Page 11 Line 451-454: “When we apply negative affective states as the independent variable, the estimated value of the indirect effect is 0.08, and the bias-corrected 95% CI of the estimated effect does not include zero [-0.13, - 0.04].” BUT it does include zero.
Ø Include VAF (variance accounted for) to confirm the strength of mediation. If VAF is less than 0.20 then it would not be considered as a mediation effect.
Ø Besides, the results seem ok.
Conclusion, implications, and limitations:
ü Discuss the results in more detail and compare them with previous studies wherever possible.
ü The paper offers some Theoretical and practical implications. There is a need to explain how this study will be useful for an international audience.
ü Limitations and future research directions are okay.
Additional comments:
v Proofreading is recommended.
v Add the line number(s) when you quote something.
v Avoid using unstandardized abbreviations such as EMS.
v All the abbreviations should be provide in a table.
Author Response
Dear reviewer,
Thank you for your constructive and valuable comments on the article. We have made revisions to all comments one by one, and have updated and improved the article based on the comments, which has greatly improved the quality of our article. The followings are our responses to your comments one by one:
Abstract
Point 1: In the first two lines, please explain the problem.
Response: Thank you for your comment! We added the explanations of the problem briefly at the beginning of the abstract (Page 1, Line 11-14): “Given the enormous value that project failure brings to individuals and organizations, a large number of scholars have explored the antecedents that affect employees’ learning from project failure. However, few scholars have paid attention to how individual’s affective states interact with cognition patterns to learn from failure.”
Point 2: Add a line on the analysis technique.
Response: Thank you for your comment! We have added analytical techniques in the abstract (Page 1, Line 17-19): “By using SPSS and Amos software, hierarchical regression analysis of questionnaire data collected from 774 employees in high-tech firms in China indicates that…”
Introduction:
Point 1: In general, it is well written. However, I would like to know what is already (if any) available in the literature on the affective states and learning relationship.
Response: Thank you for your comment! In the second paragraph of the introduction (Page 2, Line 44-62), we supplemented existing research on the relationship between affective states and learning behavior, and pointed out the research gap of existing research, as well as the differences between learning from failure behavior and learning behavior: “Usually, individuals show general daily affective states at work and different affective states usually bring about different behavioral consequences. General daily affective states can be both positive and negative. Some scholars have linked emotions or affective states to learning behavior to explore the close relationship between the two. Most scholars believe that positive emotions or affective states can promote learning while negative emotions or affective states will hinder learning. Specifically, positive affective states can push individuals to overcome a psychological dilemma after a failure faster and help them analyze problems and learn from failure. In contrast, negative affective states at work can have a significant negative impact on the learning process and learning outcomes. However, most of these studies have been conducted by psychologists or educationists, and scholars in the field of organization have yet to pay attention to the impact of employees' daily affective states on learning behavior. In addition, as a special situational event, few scholars have focused on the relationship between an individual's affective states and learning behavior after experiencing the failure or suggested appropriate mediating mechanism and boundary conditions. The relationship between daily affective states and learning from failure has been mostly viewed as common knowledge. Therefore, we add a more comprehensive set of antecedents to previous studies to investigate the specific mechanisms by which general daily affective states affect learning from failure.”
Point 2: The arguments related to the inclusion of moderating variables need to be improved.
Response: Thank you for your comment! We incorporated theoretical (i.e., cognitive behavioral theory) explanations when introducing moderating variables, providing a more detailed explanation of the reasons for selecting project commitment as the moderating variable. In addition, we have readjusted the writing logic of this paragraph to improve the inclusion of the moderating variables more thorough.
Specifically, as shown on Page 2-3, Line 79-95: “Moreover, cognitive behavior theory suggests that as a boundary condition, an individual's cognition can have an impact on emotions, affective states and behavior [16]. Specifically, an individual's positive cognition of an event will interact with their emotions or affective states to have an impact on their thinking and beliefs after experiencing the event. Previous literature has mentioned that although an individual’s general daily affective states can have impact on learning behavior, an individual’s perception of work (e.g., the perceived importance of work, organizational identity, etc.) can modify the effects of emotions [17-18]. As a positive cognition to the specific project, project commitment, in general, refers to “a person’s attachment to a project goal or determination to achieve it” [19], is a measure of how hard an individual works towards a set goal, and individuals with high levels of project commitment are not willing to easily compromise or abandon their goals even when faced with hardships and setbacks [19]. Project commitment has often been cited as an important factor influencing employee performance by providing motivation to overcome adversity and achieve goals that seem impossible or far exceed expectations [20]. We believe that project commitment can moderate the relationship between daily affective states and learning from failure by enhancing employees’ sense of responsibility and belonging.”
Point 3: Furthermore, briefly explain the theoretical implications and novelty of the study.
Response: Thank you for your valuable suggestion! At the end of the introduction, we supplemented the theoretical contributions of this article and briefly summarized the novelty of the article.
Specifically, as shown on Page 3, Line 100-110: “Our research has made contributions to existing literature in three ways. Firstly, we expand the application of cognitive behavioral theory in the field of organization research, providing important theoretical perspectives for organizational scholars to discuss the impact of the interaction between emotions and cognitions on individual behaviors. Secondly, we extend the research on the antecedents of learning from failure, creatively explore the impact mechanism of the interaction between the emotion and cognition on learning from failure. Finally, we creatively explore the important mediated mechanisms of error management strategies, shifting the previous research on error management strategies from the organization-level to the individual-level, enriching the relevant research on individual’s failure management strategies.”
Literature review:
Point 1: There should be one paragraph about Cognitive behavior theory and how it supports the proposed relationships.
Response: Based on your feedback, we do recognize that we lack an introduction to the theoretical basis of this article in the literature review. Therefore, we briefly introduced cognitive behavior theory at the end of the literature review (page 4, line 159-172).
The specific content is as follows: “In addition, cognitive behavioral theory emphasizes the role of cognitive activity in psychological or behavioral problems, which suggests that cognition plays an important role of a "coordinator" between affective states and individual behaviors[32]. In particularly, positive cognition helps individuals to perceive the effects of affective states and to take positive behaviors, while negative cognition impedes individuals to perceive the effects of affective states and to take negative behaviors[33]. We notice that in existing studies, project commitment, as an individual's positive cognition to a project, has not been received much attention from scholars. Most studies have explored the direct impact of project commitment on individuals’ positive behavior[34-35], while neglecting its crucial positive boundary role in the bond of “emotional affects—behaviors”. Therefore, we intend to include project commitment in our theoretical model to explore how this positive cognition interacts with daily affective states to have an impact on the process of affective states influencing learning from failure.”
Point 2: There are many statements/claims without the support of literature. I suggest supporting your claim in the light of literature.
Response: Thank you for your valuable advice! After sorting through the entire text, we found that we did have many statements or claims that were not supported by references. Therefore, we re-read the entire manuscript and added corresponding references after these statements/claims. We have highlighted the newly added citations in red font, which you can refer to in the revised manuscript.
Point 3: In addition, use the adopted theory in the development of the proposed hypothesis.
Response: Thank you again for your valuable advice! We have also added a series of citations and references in the hypotheses development section, and the newly added references also have been highlighted in red font, which you can refer to in the manuscript.
Point 4: Furthermore, the arguments for mediating hypotheses are not clear. More specifically, hypothesis 3b cannot be accepted because if it includes a negative value then the confidence interval would have zero. Therefore, it must be rejected.
Response: We are so apologized that we didn't understand what you meant. Hypothesis 3b describes that negative affective states will indirectly and negatively affect learning from failure through error management strategies. If this hypothesis is supported, the indirect effects tested by Bootstrap method should be negative, and the 95% confidence interval should be all less than 0 and not include 0 (i.e., all negative numbers). As we reported in the result, the indirect effect is -0.08, and the 95% confidence interval is negative and does not include zero. Therefore, hypothesis 3b should be supported.
It is common in management and organizational behavior research to explore the opposite effect of different aspects of an independent variable on the same dependent variable. We followed the research approach of most scholars and incorporated different aspects of one independent variable into the same theoretical model (e.g., positive/negative affective states), exploring the different effects of the two aspects of one independent variable (affective states) on the two different pathways of the dependent variable (e.g., learning from failure). If the indirect effect is opposite and the sign of 95% confidence interval is opposite and does not contain 0 through the Bootstrap, then the model should be supported. Some exemplary literature is Huai et al. (2022) and Sessions et al. (2020).
Of course, if we misunderstand your meaning, you can continue to point it out to us and we will accept your opinion.
References:
Huai, M., Lian, H., Farh, J. L., & Wang, H. J. (2022). Leaders’ impulsive versus strategic abuse, goal realization, and subsequent supportive behaviors: A self-regulation perspective. Journal of Management, 01492063221132481.
Sessions, H., Nahrgang, J. D., Newton, D. W., & Chamberlin, M. (2020). I’m tired of listening: The effects of supervisor appraisals of group voice on supervisor emotional exhaustion and performance. Journal of Applied Psychology, 105(6), 619-636.
Point 5: Moreover, the explanation of moderating hypotheses is not convincing and required a lot of work.
Response: Thank you for your valuable advice! We have supplemented the theoretical illustration of moderating effects and made the logic clearer. We have highlighted the specific updates in red font in the section 2.5.
Point 6: Besides, most of the cited articles are old. It should be improved with the help of recently published papers (not more than 5 years old).
Response: Thank you for your valuable advice! After reviewing the manuscript, we found that most of the cited references were old, so we updated the references to ensure that most of them were no more than 5 years. We have highlighted the specific updates in red font.
Methods, analysis, and Results:
Point 1: When the data were collected?
Response: We have added the data collection time, which can be found on Page 8, Line 362: “From March to December 2020”.
Point 2: The scales should be added in the appendix for clarity.
Response: We have added a series of appendices at the end of the manuscript, which include the scale we used, as shown on Page 22-23, Line 855-916
Point 3: What was the response rate? Did you check non-response bias?
Response: We have reported the response rate of the data in the 3.1. Participants and Data Collection section, which can be found on Page 8, Line 378. Specifically, the response rate is 14.5%. Regarding non response bias, we conducted independent sample t-tests to compare variables such as the type of companies between the participating and non-participating companies, and there was no significant difference between the two (see Table 2 in Appendix for specific results).
Point 4: Add a table and show the factor loading of each item, AVE, and CR of each scale.
Response: Based on your comment, we have added a table showing the factor loading of each item, AVE, and CR of each scale. However, due to the large size and space limitations of the table, we were unable to include it in the main text. Therefore, the table was also placed in the form of an appendix at the end of the manuscript, on Page 20-21, Line 844-847.
Point 5: Correlation is fine but I think HTMT should also be added.
Response: Based on your comment, we have added a table showing the HTMT value between each studied variable, which is shown in Table 2 on Page 21, Line 843.
Point 6: The R-square is very small? Could you please justify this?
Response: Thank you for your good question! We have some explanations for this issue. Firstly, we all know that R-square are a measure of the performance of regression models, representing the proportion of independent variables that can explain the dependent variable. In humanities and social science research, such as management and applied psychology research, due to the numerous factors that affect a certain phenomenon, selecting only a few variables to add to the regression model to predict a certain phenomenon cannot explain the entire process. Therefore, for management and social science research, most of them explore the correlation and significance level between variables. Besides, in the fields of social sciences (e.g., political science and sociology), around the mid-1980s, statisticians began to reject the R-squared model and instead accept the antecedent-consequence approach. Finding a large R-squared value can encourage the inclusion of many independent variables, and this approach may be problematic for various reasons (Achen, 2005). In short, few researchers use R-square statistics as the basis for evaluating causal model. Additionally, the R-squared of the main models in this article are all around 0.1-0.2. Compared to top journals in the fields of management and applied psychology (such as Academy of Management Journal and Journal of Applied Psychology), the R-squared values in this study are not extremely low. Below are some exemplary articles published on AMJ and JAP, where R-squared is similar to ours.
Grijalva, Maynes, T. D., Badura, K. L., & Whiting, S. W. (2020). Examining the “I” in team: A longitudinal investigation of the influence of team narcissism composition on team outcomes in the NBA. Academy of Management Journal, 63(1), 7-33.
Li, A. N., & Tangirala, S. (2022). How employees’ voice helps teams remain resilient in the face of exogenous change. Journal of Applied Psychology, 107(4), 668-692.
Point 7: Page 11 Line 451-454: “When we apply negative affective states as the independent variable, the estimated value of the indirect effect is 0.08, and the bias-corrected 95% CI of the estimated effect does not include zero [-0.13, - 0.04].” BUT it does include zero.
Response: As you mentioned above, we are so apologized that we did not understand the meaning of your comment. What we originally wrote about this sentence is: “When we apply negative effective states as the independent variable, the estimated value of the indirect effect is 0.08, and the bias corrected 95% CI of the estimated effect does not include zero [-0.13, -0.04].” The indirect effect is -0.08 instead of 0.08, so using the Bootstrap for 95% confidence interval testing should not include 0 to verify the validity of the indirect effect. And our 95% CI is [-0.13, -0.04], which is indeed all less than 0 and does not include 0, so it seems that our writing is correct.
Of course, if we misunderstand your meaning, you can continue to point it out to us and we will accept your opinion.
Point 8: Include VAF (variance accounted for) to confirm the strength of mediation. If VAF is less than 0.20 then it would not be considered as a mediation effect.
Response: As shown in Line 494-501 on Page 12, we have supplemented the VAF according to your comment. The VAF of both mediation paths is greater than 0.2, so both of our proposed mediation paths are supported.
Conclusion, implications, and limitations:
Point 1:Discuss the results in more detail and compare them with previous studies wherever possible.
Response: Thank you for your valuable comment! After reading the theoretical implication again, we found that this section was indeed described very simply, and we did not have a good dialogue between the research results and the literature. Therefore, we have rewritten the section of theoretical significance to form a communication between the research results and existing literature. The specific content is shown in Page 14-15, Line 582-642.
Point 2: The paper offers some Theoretical and practical implications. There is a need to explain how this study will be useful for an international audience.
Response: Thank you for your valuable feedback! We have added the explanations of how our research is useful for an international audience (see Page 15, Line 657-663): “Finally, we would like to say that failure knows no borders. Although our participants come from Chinese organizations, with the increasing economic and commercial uncertainty after Covid-19, failure has become an opportunity and challenge faced by various countries, societies, organizations, and individuals. Therefore, all organizations and individuals should attach importance to learning from failures and managing failure, in order to achieve long-term success.”
Additional comments:
Point 1: Proofreading is recommended.
Response: Thank you for your comment! We have carefully read and proofread the entire manuscript and corrected any errors that appeared in the manuscript.
Point 2: Add the line number(s) when you quote something.
Response: We have indicated all page numbers and line numbers in the reply letter.
Point 3: Avoid using unstandardized abbreviations such as EMS.
Response: We have removed all abbreviations in the manuscript and used full names for all variables.
Point 4: All the abbreviations should be provide in a table.
Response: We have removed all abbreviations in the manuscript and used full names for all variables. Therefore, we did not have a list of all abbreviations in a table.
Round 2
Reviewer 1 Report
Dear Authors,
The article has been completely rebuilt from the original version. The authors have adapted to the comments of the reviewers. I find that the project's goal and hypotheses have been formulated better. The article deals with a current and scientifically important topic. I have no objections to the methodology and structure of the article. Correct literature was used. The drawings were prepared correctly, and the research limitations and further research challenges were indicated.
Author Response
Dear Reviewer,
Thank you for taking the time and effort to review this article. Thank you for your very constructive and valuable feedback. With your feedback, the quality of our article has greatly improved.
Thank you again for your hard work!
Best regards.
Reviewer 2 Report
Thank you so much for inviting me to review manuscript behavsci-2406031 entitled “Are you dominated by your effects? How and when do employees’ daily affective states impact learning from project failure.” I read it with great interest. I think the author(s) have made significant efforts in revising the manuscript. I am happy with most of the changes and explanations given by the author(s). There are two concerns
1. Why the data is too old? The data collection was completed in December 2020 and now it is June 2023. As the data was collected before the start of the pandemic, I am curious whether the results are still significant after the pandemic? Could you please justify this?
2. In Appendix Table 1, there is a typo because all the CR values are below 0.5 and all AVE values are above 0.7.
Good Luck
In general, it is much better now. However, there are still some typos.
Author Response
Dear Reviewer,
Thank you for your recognition of our previous work. We also appreciate your providing us with such high-quality feedback and comments. After the last revision, we believe that the quality of our article has greatly improved. At the same time, we also appreciate your updated feedback for us. The followings are our responses to your comments one by one:
Point 1. Why the data is too old? The data collection was completed in December 2020 and now it is June 2023. As the data was collected before the start of the pandemic, I am curious whether the results are still significant after the pandemic? Could you please justify this?
Response: Thank you for raising this important question! Firstly, we acknowledge that it has been nearly two and a half years since we collected data. But we believe that our data collection time will not have a significant impact on the research results. I think there are several reasons:
First of all, you may have mistaken the time of the COVID-19 pandemic. The time when the pandemic began is early 2020. In China, the COVID-19 pandemic entered a stable period after the middle of 2020. Until now, China’s economic and social development has gradually stabilized, which will not have a major impact on our research conclusions. Therefore, our data was indeed collected after the pandemic, which is still significant for the current situation.
Secondly, we believe that COVID-19 does not have much impact on our research. Most of the studies on COVID-19 in organizational psychology have focused on topics highly related to COVID-19, such as turnover intention (Poon et al., 2022), work engagement (Liu et al., 2021; Oberländer & Bipp, 2022), negative emotions (Lades et al., 2020), negative work experience (Giauque et al., 2022), and so on. However, failure is inevitable and extremely common in organizations. Although COVID-19 may bring more failures, our research focuses on the micro psychological process that affects individual learning from failure, and the external environment will not have a great impact on our research. In order to reduce readers’ concerns, we have provided a brief explanation on the timing of data collection on Page 8, Line 371-374: “Since the data was collected after the outbreak of COVID-19 pandemic, there were no significant organizational or contextual changes in all participated organizations, which means that our data collection time will not have a significant impact on the research”.
It should be acknowledged that the daily affective states of employees may indeed be affected by COVID-19. Therefore, your comments remind us that we can further improve our research in the future study, and explore the contextual factors that affect learning from failure from the perspective of the impact of COVID-19.
References:
Giauque, D., Renard, K., Cornu, F., & Emery, Y. (2022). Engagement, exhaustion, and perceived performance of public employees before and during the COVID-19 crisis. Public Personnel Management, 51(3), 263-290.
Lades, L. K., Laffan, K., Daly, M., & Delaney, L. (2020). Daily emotional well‐being during the COVID‐19 pandemic. British Journal of Health Psychology, 25(4), 902-911.
Liu, D., Chen, Y., & Li, N. (2021). Tackling the negative impact of COVID-19 on work engagement and taking charge: A multi-study investigation of frontline health workers. Journal of Applied Psychology, 106(2), 185–198.
Oberländer, M., & Bipp, T. (2022). Do digital competencies and social support boost work engagement during the COVID-19 pandemic? Computers in Human Behavior, 130, 107172.
Poon, Y. S. R., Lin, Y. P., Griffiths, P., Yong, K. K., Seah, B., & Liaw, S. Y. (2022). A global overview of healthcare workers’ turnover intention amid COVID-19 pandemic: A systematic review with future directions. Human Resources for Health, 20(1), 1-18.
Point 2. In Appendix Table 1, there is a typo because all the CR values are below 0.5 and all AVE values are above 0.7.
Response: Thank you for your detailed review! We have indeed reversed the places of AVE and CR, and we have made corrections in the Appendix Table 1.
Point 3:In general, it is much better now. However, there are still some typos.
Response: All of authors have carefully proofed the manuscript again, confirming that the typos (i.e., words and syntax error) in the manuscript have been corrected.
Thank you again for your hard work!
Best regards.